# A Mechanism-Based Automatic Fault Diagnosis Method for Gearboxes

**DOI:** 10.3390/s22239150

**Published:** 2022-11-25

**Authors:** Lei Xu, Tiantian Wang, Jingsong Xie, Jinsong Yang, Guangjun Gao

**Affiliations:** 1School of Traffic and Transportation Engineering, Central South University, Changsha 410075, China; 2School of Mechanical and Vehicle Engineering, Hunan University, Changsha 410082, China

**Keywords:** gear faults, rotational frequency search algorithm, fault frequency identification, automatic diagnosis method

## Abstract

Convenient and fast fault diagnosis is the key to improving the service safety and maintenance efficiency of gearboxes. However, the environment and working conditions under complex service conditions are variable, and there is a lack of fault samples in engineering applications. These factors lead to difficulties in intelligent diagnosis methods based on machine learning, while traditional mechanism-based fault diagnosis requires high expertise and long time periods for the manual analysis of data. For the requirements of diagnostic convenience, an automatic fault diagnosis method for gearboxes is proposed in this paper. The method achieves accurate acquisition of rotational speed by constructing a rotational frequency search algorithm. The self-referencing characteristic frequency identification method is proposed to avoid manual signal analysis. On this basis, a framework of anti-interference automatic diagnosis is constructed to realize automatic diagnosis of gear faults. Finally, a gear fault experiment is carried out based on a high-fidelity experimental bench of bogie to verify the effectiveness of the proposed method. The proposed automatic diagnosis method does not rely on a large number of fault samples and avoids the need for diagnosis through professional knowledge, thus saving time for data analysis and promoting the application of fault diagnosis methods.

## 1. Introduction

Gearboxes have many advantages such as a constant transmission ratio, high dynamic torque, and compact structure, and they have been widely used in mechanical transmission systems in wind power generation, aviation, shipping, rail transport, and other industries [1]. Mechanical damages such as tooth surface wear, tooth surface abrasion, and tooth root cracking will lead to serious effects in terms of the transmission accuracy and service safety of gearboxes. Therefore, gearbox diagnosis for fault identification and location has a great engineering value and research significance and provides decision support for safe service and efficient maintenance.

The vibration-based fault diagnosis method has been widely used in condition monitoring systems for machines and equipment because of its convenience in obtaining extensive information related to the health status of gearboxes [2]. Yin et al. [3] proposed a statistical model of gear mesh vibration signals which compute the residual signal between the synchronous signal average and the output of the optimal signal model for efficiently detecting the gear tooth fault. Wang et al. [4] proposed an improved model for calculating the meshing stiffness of a cracked helical gear pair and investigated a vibration-based crack fault diagnosis method. Yu et al. [5] proposed an online resonant frequency identification method to extract the gear fault features in the resonant region of a planetary gearbox. Xiao et al. [6] established an elastic hydrodynamic point contact model and a wear model and analyzed the mechanical characteristics of the worn gears. Zeng et al. [7] proposed a numerical method to study the planetary gear fault spectrum mechanism. Mehrdad et al. [8] proposed parametric power spectrum analysis and support vector machines for feature extraction and classification. Zhang et al. [9] used a learning algorithm for the decomposition of singular values with optimal parameter shift to detect early faults in gearbox bearings. Zhou et al. [10] conducted a detailed study on the sinusoidal frequency modulation modes of fault-related vibration signal models. Wei et al. [11] proposed a two-stage variational linear FM component decomposition method to extract gear fault features from the Hilbert transformed amplitude modulation signal. Pan et al. [12] proposed a signal processing method for the decomposition to effectively extract the state information of the signal and apply it to the fault diagnosis of planetary gearboxes. The research on gear fault mechanisms and feature extraction methods has provided an important theoretical basis and technical means for gear fault diagnosis. However, traditional mechanism-based diagnosis requires a user’s professional knowledge related to fault mechanisms and feature extraction, and the signal analysis and diagnosis decisions are guided by professional knowledge, which requires a lot of time and is not conducive to the promotion of its use in field maintenance and inspection. Therefore, even if the mechanism-based gear fault diagnosis is more mature in both principle and method, it still lacks convenient and automatic diagnosis technology.

To overcome the reliance on user expertise, intelligent methods such as machine learning and deep learning have been introduced into fault diagnosis with the continuous development of computer technology. Singh et al. [13] proposed a new deep learning-based domain adaptation method for gearbox fault diagnosis under significant speed variations. Bangalore et al. [14] proposed an artificial neural network-based condition monitoring method applied to the fault diagnosis of gearbox bearings. Yu et al. [15] proposed a new DNN, the one-dimensional residual convolutional autoencoder, for learning features directly from vibrational signals in an unsupervised learning manner. Wang et al. [16] used principal component analysis to reduce the dimensionality of redundant statistical features and subsequently used the K-nearest neighbor algorithm to identify gear cracks under different operating conditions. Azamfar et al. [17] proposed a fault diagnosis method based on motor current signature analysis, where data obtained from multiple current sensors are fused for fault classification by a novel two-dimensional convolutional neural network architecture. Shi et al. [18] introduced a novel deep neural network based on a bidirectional convolutional long short-term memory network to determine the type, location, and direction of planetary gearbox faults. Wu et al. [19] used a long and LSTM network to capture the time-dependent features for the fault diagnosis of gearbox bearings. Yu et al. [20] proposed a new DNN model, the knowledge-based deep belief network, which inserts confidence and classification rules into the deep network structure. Huang et al. [21] proposed a residual gated dynamic sparse network to enhance the multi-sensor feature learning capability and fusion capability of gearboxes. Chen et al. [22] proposed a new diagnostic method to solve multiple types of concurrent faults to improve the fault diagnosis performance of gearboxes. Miao et al. [23] proposed a new diagnostic theory called feature mode decomposition (FMD) which adaptively and accurately decompose the fault mode. Current intelligent diagnosis methods overcome the reliance on manual analysis and decision-making of users, but the quality and quantity of fault sample data determine the merit of the model training effect. However, the complex environment and significant load variation of on-board gearboxes lead to large differences in data distribution, and it is very difficult to collect a large number of gear fault samples in the service environment.

The promotion of diagnosis technology must avoid over-reliance on the user’s knowledge, the user’s experience, and manual signal analysis, while it should not be limited by the demand and constraint of a large number of fault samples. In this paper, based on the vibration response mechanism of gearbox faults, a piece of interference-resistant automatic fault diagnosis technology is constructed. The rest of this paper consists of the following sections. The automatic diagnosis method of gear faults is proposed in Section 2. Subsequently, Section 3 presents the bogie gearbox experiment and Section 4 presents the method validation. Conclusions are shown in Section 5.

## 2. The Proposed Automatic Diagnosis Method

The proposed automatic diagnosis method of gear faults aims to improve the diagnostic convenience. The method is based on the mechanism of gearbox faults and achieves accurate acquisition of rotational frequency by constructing the rotational frequency search algorithm to overcome the inconsistency between control speed and actual speed. Furthermore, the self-referencing characteristic frequency identification method is proposed, and a framework of anti-interference automatic diagnosis is constructed.

### 2.1. Response Mechanism of Gear Faults

A pair of meshing gears can be regarded as a vibrating system with mass, spring, and damping, whose mechanical model is shown in Figure 1 and whose vibration equation can be expressed as [24]:(1)$$M\ddot{X}+C\dot{X}+K\left(t\right)X=K\left(t\right)E_{1}+K\left(t\right)E_{2}$$
where *X* is the relative displacement of the gear along the line of action; *K*(*t*) is the gear mesh stiffness; *M* is the equivalent mass of the gear pair; *E*_1_ is the average net elastic deformation of the gear after being loaded; *E*_2_ is the additional displacement caused by gear errors and failures.

The meshing stiffness *K*(*t*) of the meshing pair changes periodically with time due to the repeated process of periodically entering and exiting meshing during gear operation. Its frequency domain is the meshing frequency of the gears. If the rotational frequency of the driving wheel is *f_r_*, the number of teeth is *Z*_1_, the rotational frequency of the driven wheel is *f_r_*_2_, the number of teeth is *Z*_2_, the meshing frequency *f_m_*, and its higher harmonics can be expressed as:(2)$$f_{m}=if_{r}Z_{1}=if_{r2}Z_{2}\;,\;\;i=1,2,3$$

The vibration response component due to the meshing stiffness excitation can be expressed as:(3)$$X_{1}\left(t\right)=a\;cos\left(2\pi f_{m}t\right)$$

Taking the driving wheel gear failure as an example, the vibration response of the failed gear can be expressed as:(4)$$X_{2}\left(t\right)=a\left(1+r\cos(2\pi f_{r}t\right))\cos\left(2\pi f_{m}t\right)$$
where *r* is the amplitude modulation factor. Expanding the above equation, it can be seen that the vibration response of the faulty gear includes the frequency components of $f_{m}-f_{r}$, $f_{m}+f_{r}$ etc. In practice, there are high harmonic components in both the meshing frequency and the rotation frequency. As a result, gear faults tend to cause side frequencies at each order meshing frequency [25], as shown in Figure 2. The above failure mechanism provides important priori information for gear fault diagnosis and avoids the reliance on a large number of fault sample data.

### 2.2. The Rotational Frequency Search Algorithm

The basis of automatic diagnosis is the automatic extraction of gear characteristic frequency components, which depends on the accurate acquisition of speed signals. Considering the economy and system complexity, many machines often do not have an additional speed test system but obtain the speed signal directly from the motor control system. Due to the transmission lag and data reading priority problems, the data acquisition system has difficulty in synchronizing the vibration signal with the speed signal, resulting in a mismatch between the acquired speed signal and the tested vibration data. The rotational frequency search algorithm is proposed for the problem of inaccurate rotational frequency. It takes advantage of the inter-constrained relationship among the frequency components such as the rotational frequency component, the meshing frequency component and its higher harmonics in the gear rotor system. The flow of the algorithm is shown in Figure 3 and its main steps are as follows.

(1)The maximum time lag ($\Delta t=\Delta t_{1}+\Delta t_{2}$) is calculated based on the parameters of transmission and interaction. Combined with the maximum acceleration (a) of the device and the rotational frequency (acquired from CAN data bus or displayed by the controller), the frequency range ($\left\lceil f_{l},f_{h}\right\rceil$) of the true rotational frequency can be calculated and the frequency range of the higher order harmonics ($2\times \lceil f_{l},f_{h}\rceil$, $3\times \lceil f_{l},f_{h}\rceil$ etc.) can be inferred.(2)Spectrums of the acquired vibration signal are calculated and the upper envelope curve on the spectrum in the range of the concerned higher order harmonics is extracted.(3)The frequencies of all envelope peaks are extracted and compared with the frequency bands of rotational frequencies and each higher order harmonics. The frequencies of peaks within the frequency band are recorded as rotational frequencies or their higher harmonics.(4)The extracted higher order harmonics are used to extrapolate the predicted rotational frequency.(5)Based on the predicted rotational frequency, the frequency band of the gear-mesh is calculated and the meshing frequency is extracted as a corroboration frequency.(6)The rotational frequency inferred from the meshing frequencies is finally used as the searched rotational frequency.

Using the rotational frequency search algorithm, the accurate rotational frequency can be obtained, and the synchronization problem between the displayed speed and true speed corresponding to the collected vibration signal can be overcome.

### 2.3. Self-Referencing Characteristic Frequency Identification Method

The traditional mechanism-based fault diagnosis needs manual analysis of vibration signals to observe whether there are obvious fault characteristics in the signal waveform or spectrum. However, manual observation of features such as waveforms and spectra rely heavily on the experience and expertise of the user and requires significant data analysis time. This section constructs a self-referencing feature frequency identification method based on the gear fault response mechanism to realize automatic identification of fault feature frequency components in gear vibration signals.

In the manual analysis of the signal, the user can accurately and quickly determine whether there is a feature frequency component of the fault by observing the significance of the spectral peak at the feature frequency in the spectrum. Inspired by the intuition of visual senses in the manual signal analysis, the peak outlier index (*PI*) and spectral peak saliency morphology index (*SI*) are constructed from two visual attention elements to replace the manual signal analysis. The main principle of the method is shown in Figure 4.

*PI* is constructed to evaluate whether the peak is significantly larger than the amplitude of the surrounding band by the ratio of the peak at the fault feature frequency to the mean value of the spectral line within the band of interest. The indicators are constructed as follows:(5)$$PI_{i}=\frac{A_{i}}{\frac{1}{N}\sum_{j=1}^{N}s_{j}}$$
where $PI_{i}$ indicates the peak indicator at the *i-th* fault characteristic frequency, $A_{i}$ indicates the peak of the spectral line at the *i*-th fault characteristic frequency, *N* indicates the number of spectral lines in the band of interest, and $s_{j}$ indicates the amplitude of the spectral line.

*SI* evaluates the significance of the spectral peak pattern by the ratio of the peak at the characteristic frequency of the fault to the peaks at its two discrete frequency points on the left and right sides. The index is constructed as:(6)$$SI_{i}=\frac{A_{i}}{s_{i-1}+s_{i+1}}$$
where $SI_{i}$ indicates the significance index of the pulse pattern at the *i*-th fault characteristic frequency, $A_{i}$ indicates the peak of the spectrum at the *i*-th fault characteristic frequency, and $s_{i-1}$ and $S_{i+1}$ indicates the amplitude of the two frequency points on the left and right side of the fault characteristic frequency.

Based on *PI* and *SI*, the method for identifying the frequency components of faults is constructed as:(7)$$CI_{i}=\alpha PI_{i}+\beta SI_{i}\geq \;\lambda$$
where when the fusion index ($CI_{i}$) is greater than λ, it is considered that there is a significant harmonic component at the characteristic frequency. The parameters α, β, and λ, can be determined empirically by the designer or calibrated by several sets of typical fault spectra, whenever the parameters obtained can make the difference between the characteristic frequencies of normal and faulty gears significant enough.

$CI_{i}$ is constructed by signal self-reference information, avoiding the disadvantages that thresholds are disturbed by environmental and operating conditions. At the same time, it replaces the visual sense of manual analysis to achieve the frequency identification of fault characteristics, avoiding the dependence on the user’s experience and significantly reducing the signal analysis time.

### 2.4. The Framework of Anti-Interference Automatic Diagnosis

The framework of anti-interference automatic diagnosis for gear faults is constructed based on the rotational frequency search algorithm and the self-referencing characteristic frequency identification method, as shown in Figure 5. This diagnostic framework adopts multi-channel and multi-frequency joint diagnosis mechanisms, and the diagnostic interference caused by the single frequency component, calculation errors, or individual channel abnormality can be avoided.

The proposed framework of anti-interference automatic diagnosis technology mainly consists of two parts: automatic signal analysis and automatic fault diagnosis logic. The signal analysis module aims to replace the manual data analysis and identify the fault characteristic frequency components from the vibration data based on the fault mechanism. The fault diagnosis logic module is designed to replace the manual decision-making process by using the occurrence of fault characteristics frequency.

Using this diagnosis framework as a guide, an automatic diagnosis flow and logic can be constructed as follows.

(1)Calculate the gear ratio using structural parameters such as the gear pair meshing relationship and the number of teeth.(2)Obtain the displayed rotational speed signal (or controller set rotational speed) and the sampling frequency and determine the exact rotational speed and frequency of the input axis using the rotational frequency search algorithm.(3)Combining the gear ratio and the obtained rotational frequency of the input shaft, the rotational frequency of each gear shaft and the meshing frequency of each meshing pair are calculated.(4)Eight fault characteristic frequencies such as first-order and second-order side frequencies of the first- and second-order mesh frequencies on the left and right (as shown in Figure 2) are involved in the fault decision.(5)The above eight fault characteristic frequency components are identified using the self-referencing feature frequency identification method to determine whether there is a fault component at the corresponding frequency.(6)The identification results of fault characteristic frequency components are analyzed, and fault diagnosis decisions are made by using the logic shown in the framework of anti-interference automatic diagnosis.

If there is no characteristic frequency, the diagnosis results is “no fault”.

If the counts of fault characteristic frequency components in one channel exceed the threshold at a certain speed, the diagnosis results is “fault indication”.

If the counts of fault characteristic frequency components in one channel at different speeds or in multiple channels all exceed the threshold, the diagnosis results is “gear fault location”, and the faulty gear is located based on the characteristic frequency and the fault mechanism.

Engineering users can obtain the diagnosis results of gears directly using this diagnostic framework and method without manually analyzing the vibration signals. This method saves a certain amount of analysis and diagnostic time and it does not depend on the user’s experience and expertise in data analysis and does not require various types of fault samples.

## 3. The Bogie Gearbox Experiment

A high-fidelity fault simulation experimental bench of bogie is used to validate the proposed automatic fault diagnosis method of the gear. The following section will introduce the experimental setup and experimental process.

### 3.1. Experimental Setup

#### 3.1.1. High-Fidelity Experiment Bench of Bogie

According to the real structure of the bogie, a high-fidelity fault simulation experimental bench of bogie was designed and established. The combined units of the experimental bench are shown in Figure 6a, including the experimental bench main body, lubrication system, control system, and load system. The main body of the experimental bench is shown in Figure 6b, which consists of the bogie frame, drive motor, transmission gearbox, axle, supporting hybrid bearing, axle box, and base.

The control system realizes the integrated control of the experimental bench, including motor speed control, load torque control, etc., which can realize the simulation of multi-speed and variable torque load. The speed range of the test stand motor is 300–3600 r/min. It can make the axle reach a speed of 125–1500 r/min, which is consistent with the actual bogie speed range. The magnetic powder brakes on both ends of the spindle can apply a braking torque from 0 to 50 N m. Bearings and drive gears can be quickly disassembled to enable simulation studies of each typical fault. In summary, the experimental bench has similar composition, similar transmission structure, similar boundary, and consistent load characteristics to the real bogie, which can effectively realize the verification of the diagnosis method.

#### 3.1.2. Acquisition System of Vibrations

The acquisition system of vibration data is shown in Figure 7, which mainly contains vibration sensors, a data collector, an acquisition control, and data analysis systems. The parameters of the sensors and collectors used in this experiment are shown in Table 1. The acquisition control and data analysis system can realize the setting of acquisition parameters, acquisition control, data display, data storage, data analysis, and visualization.

### 3.2. Experimental Process

Based on the bogie gearbox experiment system, gear fault experiments are carried out to obtain vibration data of different gear faults and to verify the proposed fault diagnosis method. It mainly includes fault parts preparation and measurement point arrangement.

#### 3.2.1. Fault Parts Preparation

Pinion gears of gear meshing pairs in engineering are more prone to failure than large gears due to frequent load alternations. In this experiment, the pinion gear is used as the test object to produce typical gear damage, including normal, scratch, tip broken, and root crack, as shown in Figure 8. Scratch damage is made by manual contusion. Root crack damage is made by wire cutting. The crack width is 0.2 mm and the crack depth is 30% of the tooth root width. The tip broken damage is machined by milling, where the milling width is 1/3 of the top width.

#### 3.2.2. Measurement Point Arrangement

The vibration acceleration sensors are arranged at three locations, as shown in Figure 9. Figure 9a shows the gear for the experiment and the faulty gear is the pinion. The pinion shaft is the output shaft of the motor and the shaft of the large gear is the axle of bogie. Figure 9b shows a vibration acceleration sensor which is installed in the plane of the box near the pinion to obtain the pinion vibration response data. Figure 9c shows the rotating arms where the large gear is located. The vibration characteristics of the pinion are transmitted directly to the structure through the large gear and the axle. Therefore, two vibration acceleration sensors are installed at each end of the large gear shaft (the position of the rotating arm), as shown in Figure 9d.

## 4. Experiment Results and Method Validation

In this paper, a degradation stage recognition method of bearings is proposed based on outlier cleaning. The outlier detection method combining global abnormal segments detection and accurate location of abnormal impulses is constructed, realizing the accurate and quick removal of impulse-types outliers which have significant interference with the identification of degradation points. The main conclusions of this paper are as follows:

The mechanism-based manual diagnosis method and the proposed mechanism-based automatic diagnosis method are compared to verify the feasibility and superiority of the automatic diagnosis method.

### 4.1. Manual Diagnosis Method

Manual diagnosis means that the presence or absence of a fault is determined by observing at the characteristic frequency whether there exists a significant fault frequency component.

In the experiment, the motor speed was set to 1000 RPM by the controller, but there was an error between the actual output speed of the motor and the control speed. The number of pinion teeth for the drive gear is 21 and the number of teeth for the large gear is 75. Therefore, the rotational frequency of the pinion shaft (*fr*) is about 16.6 Hz and the gear meshing frequency (*fm*) is about 348 Hz. According to the mechanism analysis, the pinion fault is characterized by the side frequency components of the pinion shaft rotational frequency on both sides of the meshing frequency.

Spectrums of the acquired vibration signals are shown in Figure 10, where Figure 10a corresponds to the measured point of the axle box rotating arm at the driving end, Figure 10b corresponds to the measured point of the axle box rotating arm at the non-driving end, and Figure 10c,d are the local enlargement of Figure 10a,b.

As shown in Figure 10, the amplitude of the component at the meshing frequency (*fm*) of normal gears is low and there are no significant side frequencies on either side of the meshing frequency, i.e., *fm + fr*, *fm + 2fr*, *fm − fr*, *fm − 2fr*. On the contrary, for the three states of the faulty gears, it can be seen that there are significant components at meshing frequencies and there are intuitive and significant frequency components at the faulty characteristic frequencies (*fm + fr*, *fm + 2fr*, *fm − fr*, *fm − 2fr*) of the pinion. Therefore, the status determination of normal or faulty gears can be realized by using the vibration data of normal gears as a reference and manually featuring the frequency amplitude comparison.

### 4.2. Automatic Diagnosis Method

In order to avoid the manual spectrum analysis in the above diagnosis process, the proposed automatic diagnosis method is conducted to realize the automatic diagnosis of faults.

#### 4.2.1. Rotational Frequency Search

Firstly, the rotational frequency search method is verified. In this case, the controller shows the rotational speed as 1000 RPM, and there is an error between the actual speed of the motor and the speed set by the controller. Assuming that there is ±40 RPM error in the measured speed, the minimum speed obtained is 960 RPM and the maximum speed is 1040 RPM. The estimated rotational frequency range is from 16.0 Hz–17.3 Hz.

The results of the rotational frequency search are given in Figure 11 and Table 2. Four peaks can be extracted in the envelope curve shown in Figure 11. By comparing the rotational frequency and its higher order harmonics, it can be determined that 75.2 Hz does not belong to the rotational frequency or any higher order harmonics, while 16.1 Hz, 50.2 Hz, and 97.3 Hz are within the frequency band of 1×, 3×, and 6× harmonics, respectively.

In order to avoid the fluctuation of rotational speed and the frequency calculation error in the low frequency band, the higher mesh frequency is used as the corroboration frequency to finally infer the rotational frequency. The average rotational frequency calculated by the three searched frequencies is 16.4 Hz. The mesh frequency was then inferred to be 344.4 Hz by the mesh relationship. The mesh frequency extracted in the frequency band range around the inferred mesh frequency (344.4 Hz) is 343.11 Hz, as shown in Figure 10 and Table 2. Finally, the number of pinion teeth, 21, is used to deduce the inferred rotational frequency (343.11 Hz/21 = 16.3 Hz), which corresponds to 978 RPM.

The search process of the rotational frequency is automatically executed by the diagnostic program, without manual participation in data analysis.

#### 4.2.2. Characteristic Frequency Identification

After the rotational frequency is obtained, the characteristic indexes *PI* and *SI* for frequency component identification can be extracted at the four fault characteristic frequency components (*fm + fr*, *fm + 2fr*, *fm − fr*, *fm − 2fr*) on both sides of the meshing frequency, which are Fre1, Fre2, Fre3, and Fre4 in Table 3 and Table 4.

According to the self-referencing characteristic frequency identification method in Section 2.3, *PI* and *SI* are calculated for the drive end of normal gear and the scratch gear. Then, we kept adjusting their weight parameters to make the difference between the normal and faulty feature frequencies greater and further selected the threshold lambda accordingly. In this example, after a certain amount of parameter combinations, we eventually selected the parameters alpha and beta as 0.8 and 0.2 which makes the difference the most significant. We took α=0.8, β=0.2 and used the formula ($CI_{i}=\alpha PI_{i}+\beta SI_{i}$) to calculate CI at each characteristic frequency, as shown in Table 3 and Table 4.

By comparing Table 3 and Table 4, it can be seen that the value of CI of normal gears is less than two, while the value of CI of scratch gears is greater than two. Therefore, these two samples can be used as a benchmark and the critical value can be set to λ=2. Finally, all characteristic frequencies of the drive side for normal gears are determined to be free of the fault characteristic frequency component, and their status is marked with “0” in Table 3 (CI<λ). All characteristic frequencies at the drive end of the scuffed gear are determined to have the fault characteristic frequency component, and the status is marked with “1” in Table 4 (CI>λ).

#### 4.2.3. Fault Localization

Based on the thresholds and fusion coefficients calibrated by the normal and scuffed samples at the drive end, the vibration acceleration signals of the gears in four states of normal, scratch, tip broken, and root crack in two measurement points can be discriminated.

There are eight characteristic frequencies in total in the signals from two measuring points of each faulty gear. When a fault characteristic frequency component is determined to exist at a certain frequency (included in the eight characteristic frequencies), the count increases by one. The cumulative results of each measuring point are shown in Table 5 (the third column). From the counts of characteristic frequency components, it can be seen that only one characteristic frequency component is identified in two measurement points of the normal gear, while the characteristic frequency component counts of the faulty gear are greater than seven.

According to the fault diagnosis technology framework in Section 2.4, the warning threshold of the measurement point is set to two. When the fault characteristic frequency component counts of the measurement point exceed two, the channel is determined to be abnormal. When an abnormality occurs in a single measurement point, early warning of that fault is performed. When two or more measurement points are abnormal, the fault is determined and the faulty gear is located at the gear which the characteristic frequency belongs to.

In this case, the characteristic frequency counts of the two measurement points for the normal gear are less than two, so there is no “abnormal” state in the “measurement point warning” and the final fault identification status is “normal pinion”. For the three fault states of the pinion (scratch, tip broken, and root crack), the characteristic frequency counts of both measurement points are greater than two. Therefore, both measurement points are “abnormal”. According to the technical framework of fault diagnosis, it can be determined that there is a gear fault. The counts of fault characteristic frequency components correspond to the pinion, so the fault can be localized to the pinion gear.

#### 4.2.4. Comparison and Verification of the Results

In this example, the proposed automatic fault diagnosis method is validated with gears of four states. The comparison and verification of the results show that:

The proposed rotational frequency search algorithm can accurately extract the actual rotational speed of the gear shaft in the case of inaccurately measured rotational speed, which provides the basis for characteristic frequency identification.

The proposed self-referencing characteristic frequency identification method can accurately determine whether there is a significant frequency component at a certain frequency to achieve automatic discrimination of fault frequency components.

Faults are accurately identified and located based on the framework of anti-interference automatic diagnosis technology, which is constructed by frequency component identification, frequency component counting, measurement point warning, and fault identification.

Therefore, the above validation results show that the proposed method can achieve automatic diagnosis of gear faults without the need for extensive manual data analysis and without relying on a large amount of sample data.

## 5. Conclusions

To improve the convenience of gear diagnosis, an automatic gearbox fault diagnosis method is proposed in this paper. The effectiveness of the proposed method is verified via a bogie experiment with faulty gears. The main conclusions are as follows.

(1)The rotational frequency search algorithm, which is constructed to overcome the problem of inconsistency between the control speed/feedback speed and the real-time speed of sampled vibration data, realizes the accurate calculation of rotational frequency and provides a basis for the automatic characteristic frequency identification.(2)A self-referencing characteristic frequency identification method is proposed, and the fault characteristic frequency components are identified automatically. The indicators (*PI* and *SI*) are constructed to simulate the manual spectrogram identification process. It avoids the problem that the way of discriminating frequency components is by an absolute threshold, which is vulnerable to noise and working conditions’ interference.(3)The framework of anti-interference automatic diagnosis is constructed to realize automatic extraction of fault features based on a rotational frequency search algorithm and a self-referencing characteristic frequency identification method. The diagnostic interference caused by a single frequency component, calculation errors, or individual channel abnormality can be avoided by a multi-channel and multi-frequency joint diagnosis mechanism.

## Figures and Tables

**Figure 1 sensors-22-09150-f001:**
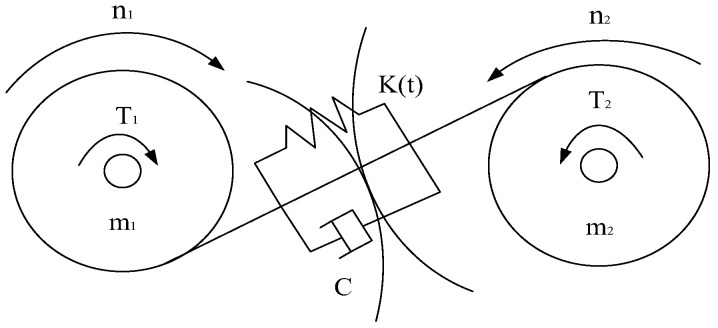
Gear meshing mechanics model.

**Figure 2 sensors-22-09150-f002:**
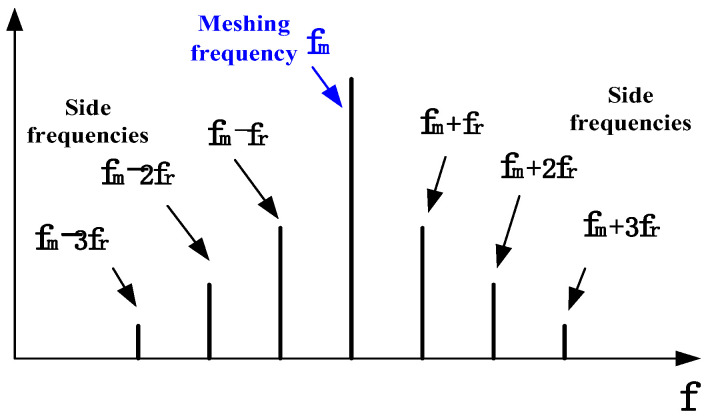
Typical frequency characteristics of gear faults.

**Figure 3 sensors-22-09150-f003:**
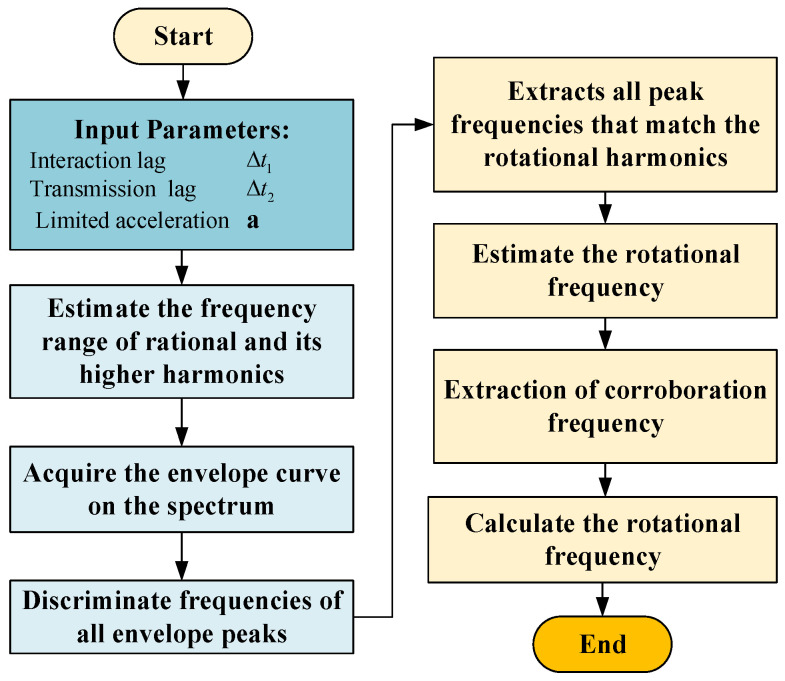
The flows of rotational search algorithm.

**Figure 4 sensors-22-09150-f004:**
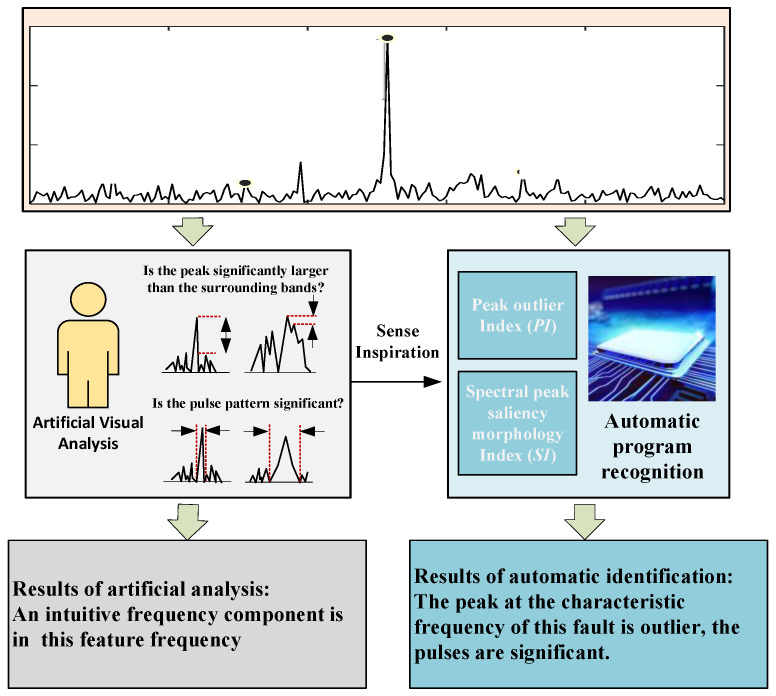
Principle of self-referencing characteristic frequency identification method.

**Figure 5 sensors-22-09150-f005:**
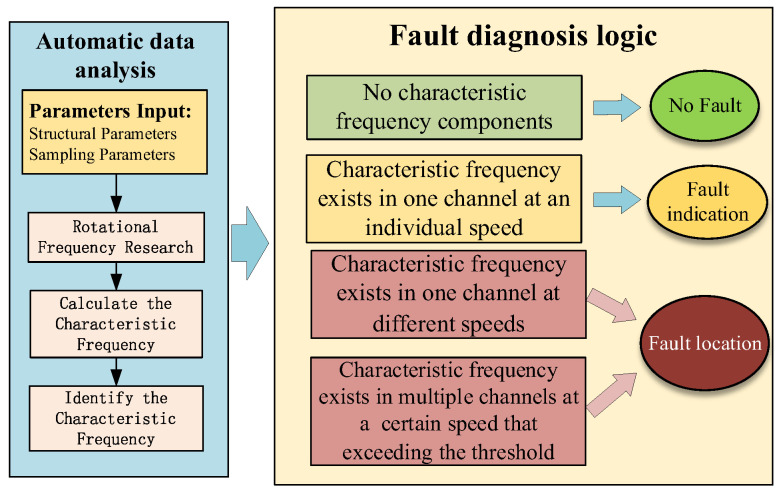
Framework of anti-interference automatic diagnosis.

**Figure 6 sensors-22-09150-f006:**
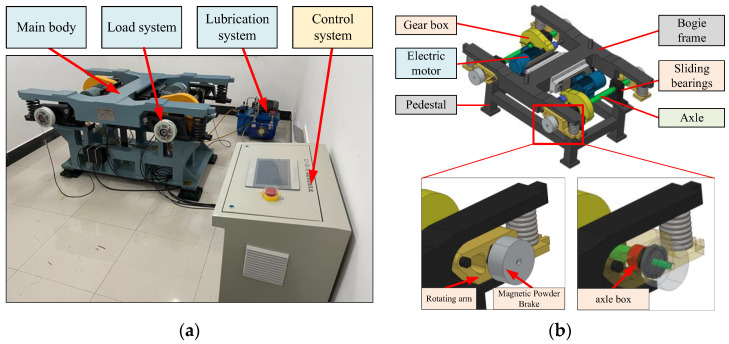
High-fidelity fault simulation experimental bench of bogie: (**a**) experimental bench; (**b**) main body of the experimental bench.

**Figure 7 sensors-22-09150-f007:**
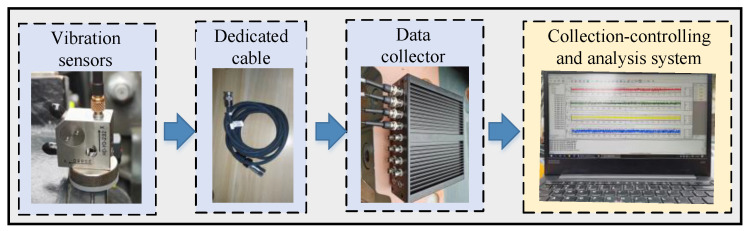
Acquisition system of vibration data.

**Figure 8 sensors-22-09150-f008:**
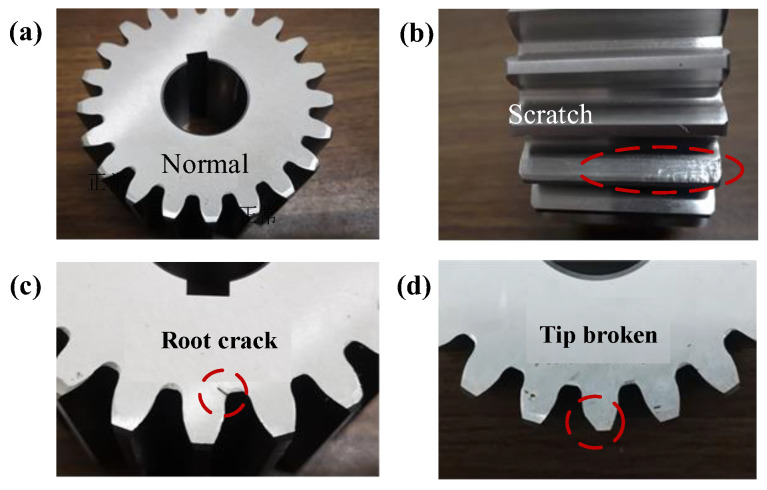
Gear fault specimen: (**a**) normal; (**b**) scratch; (**c**) root crack; (**d**) tip broken.

**Figure 9 sensors-22-09150-f009:**
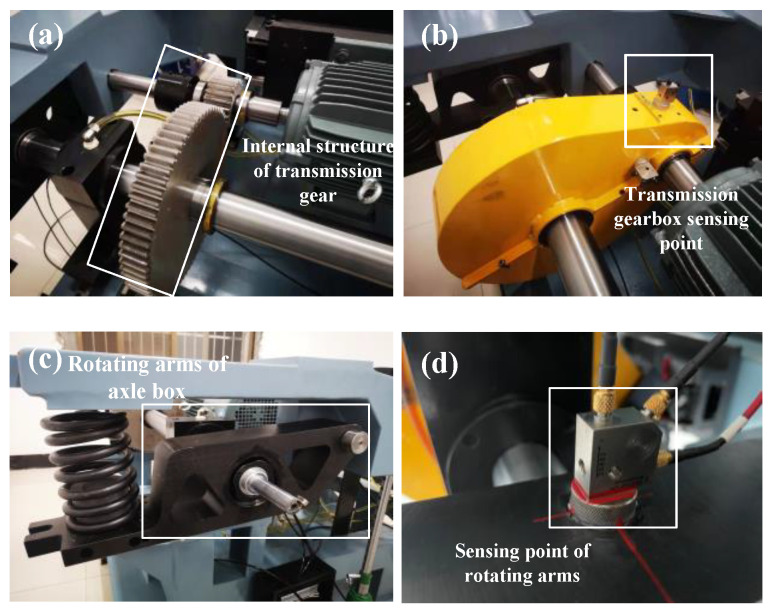
Vibration acceleration measurement point arrangement: (**a**) Internal structure of transmission gear; (**b**)Transmission gearbox sensing point; (**c**) Rotating arms of axle box; (**d**) Sensing point of rotating arms.

**Figure 10 sensors-22-09150-f010:**
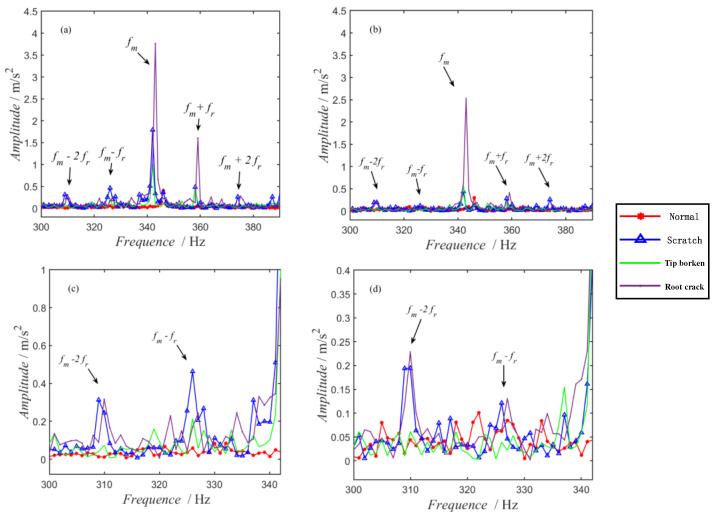
Spectrums of acceleration: (**a**) measured points at the driving end; (**b**) measured points at the non-driving end; (**c**) local enlargement of (**a**); (**d**) local enlargement of (**b**).

**Figure 11 sensors-22-09150-f011:**
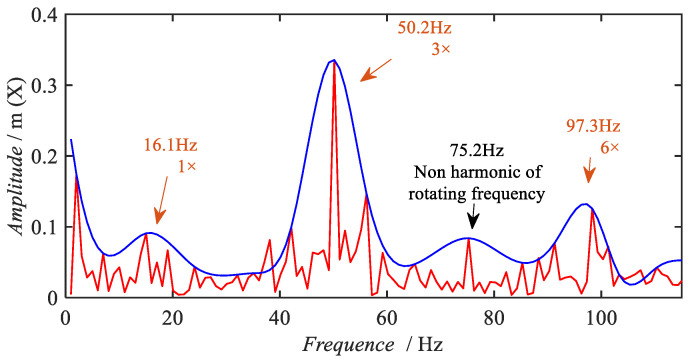
Peak envelope of the acceleration spectrum.

**Table 1 sensors-22-09150-t001:** Parameters of the sensors and collectors.

Sensitivity/(mV/g) (20 ± 5 °C)	Measurement Range/g (Peak)	Sampling Accuracy/Bit (Simultaneous Sampling)	Max Sampling Frequency/(KS/s)	Signal to Noise Ratio/(Db)
100	±50	16	102.4	96

**Table 2 sensors-22-09150-t002:** Rotational frequency search identification parameters and results.

	Estimated Rotational Frequency Range/Hz	Active Search/Hz	Corroboration Frequency/Hz	Inferred Rotational Frequency/Hz
Frequency value	16.0–17.3	16.1, 50.2, 97.3	343.1	16.3
Physical meaning	1× possible range	1×, 3×, 6×	Mesh frequency	Precise 1×

**Table 3 sensors-22-09150-t003:** Identification results of the measurement point at the drive end of the normal gear.

	Fre1	Fre2	Fre3	Fre4
*PI*	1.401	1.627	1.371	1.877
*SI*	1.744	2.300	1.552	2.074
*CI*	1.470	1.761	1.408	1.917
*λ*	2.000			
Discriminatory results (0/1)	0	0	0	0

**Table 4 sensors-22-09150-t004:** Identification results of the measurement point at the drive end of the scuffed gear.

	Fre1	Fre2	Fre3	Fre4
*PI*	2.347	2.0980	3.320	3.503
*SI*	3.082	2.0337	18.996	7.696
*CI*	2.494	2.085	6.455	4.3419
*λ*	2.000			
Discriminatory results (0/1)	1	1	1	1

**Table 5 sensors-22-09150-t005:** Pinion Gear Fault Identification.

		Characteristic Frequency Counting	Early Warning Threshold	Measurement Point Warning	Fault Identification and Localization
Normal	Drive end	0	2	Normal	Normal pinion
	Non-driver end	1		Normal	
Scratch	Drive end	4		Abnormal	Pinion fault
	Non-driver end	4		Abnormal	
Tip broken	Drive end	4		Abnormal	Pinion fault
	Non-driver end	3		Abnormal	
Root crack	Drive end	4		Abnormal	Pinion fault
	Non-driver end	3		Abnormal	

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
