# Peer review of "A Mechanism-Based Automatic Fault Diagnosis Method for Gearboxes"

_sensors, 2022, doi:10.3390/s22239150_

Round 1

Reviewer 1 Report

An automatic fault diagnosis method for gearboxes is proposed in this paper. The method achieves accurate acquisition of rotational speed by constructing a rotational frequency search algorithm. The self-referencing characteristic frequency identification method is proposed to avoid the manual signal analysis. The proposed automatic diagnosis method does not rely on a large number of fault samples and avoids the needing of diagnosis professional knowledge, saving time for data analysis and promoting the application of fault diagnosis methods.

The research results of this paper are of great significance for gear diagnosis. The the results are accurate, and the demonstration is sufficient, meeting the requirements of the journal.

There are several minor problems in the current version:

1) The fault mechanism in Section 2.1 is not the contributions of this paper and should be simplify;

2) In Section 2.2, the second step of the algorithm:”extract the upper envelope curve on the spectrum”. How to calculate it?

3) How to set the  in Formula 7? It should be supplemented.

4) How does this method diagnose multiple gear faults? Please explain.

5) There is a wrong word in Figure 5. The full text should be carefully checked.

Author Response

The authors are grateful to receive comments from Reviewer #1. Thank you very much for giving such detailed suggestions for changes. A major rewriting of the paper has been undertaken and responses to the comments are shown as follows.

Reviewer 2 Report

In this paper, an automatic gearbox fault diagnosis method is proposed. The topic is interesting. However, following issues should be addressed.

1.    In Eq. (7), more details should be added to explain how to determine parameters.

2.    In general, the meshing frequency is obvious in the spectrum of the vibration signal of healthy gear. However, from Fig. 10, the meshing frequency cannot be found.

3.     Based on Table 5, it is difficult to determine the fault type of gear.

Author Response

The authors are grateful to receive comments from Reviewer #2. Thank you very much for giving such detailed suggestions for changes. A major rewriting of the paper has been undertaken and responses to the comments are shown as follows.

Reviewer 3 Report

To improve the convenience of gear diagnosis, an automatic gearbox fault diagnosis method is proposed in this paper. The effectiveness of the proposed method is verified via a bogie experiment with fault gears. The research in this paper has a certain inspiration for the research of automatic diagnosis methods. However, the following problems in the promotion and application of the research results need to be explained:

1. Is this method applicable in case of large speed fluctuation? How to adjust parameters?

2. Can the automatic diagnosis method be used in bearing diagnosis? What changes should be make?

3. Several related articles should be quoted.

4. Some section titles in the text are error and should be carefully corrected, such as, section 4.2.1, section 4.2.3,section 3.2.1.

5. The format of the table does not meet the requirements, and it should be: TABLE+title, above the table.

Author Response

The authors are grateful to receive comments from Reviewer #3. Thank you very much for giving such detailed suggestions for changes. A major rewriting of the paper has been undertaken and responses to the comments are shown as follows.

Round 2

Reviewer 2 Report

The revised paper has properly addressed my comments.